

# Local ancestry prediction with *PyLAE*

Nikita Moshkov[1,2,3,4,*], Aleksandr Smetanin[5,*] and
Tatiana V. Tatarinova[6,7,8,9]

[1] Doctoral School of Interdisciplinary Medicine, University of Szeged, Szeged, Hungary
[2] Synthetic and Systems Biology Unit, Biological Research Centre, Szeged, Hungary
[3] Atlas Biomed Group Limited, London, United Kingdom
[4] Laboratory on AI for Computational Biology, Faculty of Computer Science, HSE University, Moscow, Russia
[5] ITMO University, Saint-Petersburg, Russia
[6] Department of Biology, University of La Verne, La Verne, CA, United States
[7] Siberian Federal University, Krasnoyarsk, Russia
[8] Institute of General Genetics, Moscow, Russia, Moscow, Russia
[9] Institute for Information Transmission Problems, Moscow, Russia, Moscow, Russia
* These authors contributed equally to this work.

Corresponding author
Tatiana V. Tatarinova,
ttatarinova@laverne.edu

## ABSTRACT

**Summary:** We developed *PyLAE*, a new tool for determining local ancestry along a genome using whole-genome sequencing data or high-density genotyping experiments. PyLAE can process an arbitrarily large number of ancestral populations (with or without an informative prior). Since *PyLAE* does not involve estimating many parameters, it can process thousands of genomes within a day. *PyLAE* can run on phased or unphased genomic data. We have shown how *PyLAE* can be applied to the identification of differentially enriched pathways between populations. The local ancestry approach results in higher enrichment scores compared to whole-genome approaches. We benchmarked PyLAE using the 1000 Genomes dataset, comparing the aggregated predictions with the global admixture results and the current gold standard program RFMix. Computational efficiency, minimal requirements for data pre-processing, straightforward presentation of results, and ease of installation make *PyLAE* a valuable tool to study admixed populations.
**Availability and implementation:** The source code and installation manual are available at https://github.com/smetam/pylae.

## INTRODUCTION

In association studies, researchers combine samples with different (usually opposing) phenotypes and compare Single Nucleotide Polymorphisms (SNPs) frequencies in two groups. When enough samples are available (typically, thousands for complex traits), a significant difference in frequencies between the groups suggests an association between the position on the genome and the studied phenotype. However, there is a possibility that the association is due to inhomogeneity of the study group in terms of provenance/ origin (for example, all people with the disease are of French origin, and the healthy cohort is Bulgarian). In this case, two populations may have different frequencies of ancestry informative markers (AIM) that are not causal to the phenotype.

An intuitive approach to solving this problem is determining the population structure first and then adjusting for it. However, this strategy is complicated for admixed populations. Due to meiotic recombination during transmission of genetic material to the offspring, individual segments of the genome may have different origins. Recombination makes it hard to determine the source of such individuals (or plant populations), especially locus-specific local origin.

Several solutions to this problem exist, such as LAMP, LAMP-ANC, and RFMIX. However, there is always room to develop a user-friendly, fast, accurate approach. The LAMP (*Sankararaman et al., 2008*) algorithm determines local origin in mixed populations. The LAMP input includes the recombination rate, global mixing ratio, and the number of generations that have passed since the mixing started. The recombination rate can be considered known due to previous studies (*Nachman & Crowell, 2000*), while the global proportion can be estimated using, for example, the ADMIXTURE tool (*Alexander, Novembre & Lange, 2009*). The idea of the LAMP method is as follows. The iterated conditional modes clustering algorithm (ICM) determines the likelihood that a genome segment within a selected window has a specific origin. An individual SNP is assigned the source by the "popular vote" approach using the inferred origins of all windows covering this position. ICM is a modification of the Expectation-Maximization (EM) algorithm, which receives a point estimate at the E-step under the assumption that the *a priori* estimates are sufficiently accurate. Therefore, ICM is faster than the EM approach. To construct an accurate *a priori* estimate, LAMP uses the MAXVAR algorithm, which works only for two populations. The size of the genomic window is selected to minimize classification errors. LAMP works fast, but its accuracy decreases if the ethnic admixture ratio is close to 1:1.

LAMP-ANC (*Pasaniuc et al., 2009*) is a modification of the LAMP tool, showing a higher accuracy than LAMP. This modification also allows triple mixing to be estimated, while LAMP cannot determine frequencies for more than two ancestral populations.

The Machine Learning algorithm RFMIX (*Maples et al., 2013*; *Uren, Hoal & Möller, 2020*) treats origin as a hidden parameter in its statistical model. RFMIX employs conditional random fields and decision trees. The RFMIX model does not impose restrictions on the number of ancestral populations and types of mixing.

We developed *PyLAE*-a new tool for determining local origin along a genome using whole-genome sequencing data or high-density genotyping experiments. *PyLAE* can process an arbitrarily large number of ancestral populations (with or without an informative prior). *PyLAE* does not involve the estimation of many parameters. *PyLAE* is a valuable tool to study admixed populations (*Hester et al., 2011*; *N'Diaye et al., 2011*; *Kantor et al., 2013*). We have tested this approach using the 1000 Genomes database.

## MATERIALS AND METHODS

### Data source

Whole-genome sequencing data in the VCF format were obtained from the 1000 Genomes Project (ftp://ftp.1000genomes.ebi.ac.uk/vol1/ftp/, genome version GRCh37). We used the

dataset of 2,504 worldwide individuals with the reported ethnic origin (*Sudmant et al., 2015*) and phased genome sequence data.

## Data pre-processing

### Admixture

To confirm the reported origin, we have extracted 130,000 ancestry informative markers identified by a worldwide study contacted by the National Genographic consortium, as described in (*Elhaik et al., 2014*). The supervised admixture analysis was performed using the K = 9 component division of ancestral populations into the following categories: North-East Asian, Mediterranean, South African, South-West Asian, Native American, Oceanian, Southeast Asian, Northern European, and Sub-Saharan African components. We used the putative ancestral populations from our earlier study (*Elhaik et al., 2014*). The admixture components represent proportions of an individual's genotype attributed to each of the nine putative ancestral genomes. The obtained nine-dimensional vectors were clustered based on the L2 norm (Euclidean distance). The optimal number of clusters was determined using the weighted Kullback-Leibler distance approach (*Tatarinova, Bouck & Schumitzky, 2008*; *Tatarinova & Schumitzky, 2014*). Within each group, the admixture profiles of individuals are similar. This dimension-reduction step allows the identification of potentially admixed individuals. This assignment was validated using haplogroup information and reAdmix analysis in group mode (*Kozlov et al., 2015*). reAdmix algorithm represents an individual as a weighted sum of present-day populations (*e.g.*, 50% British, 25% Russian, 25% Han Chinese) based on K admixture components. Instead of solving an "exact admixture" problem, we aim to find the smallest subset of populations whose combined admixture components are close to those of the individuals within a small tolerance margin. Due to the range of natural variation, the admixture proportions can be considered exact neither for the reference populations nor for the tested individuals. The admixture proportions we use are merely maximum likelihood estimates and may fail to be exactly equal to the actual shares of ancestral genomes. Therefore, determining ethnic proportions is mathematically and computationally more challenging than finding a single most fitting bio-origin. From our experience with reAdmix, this tool accurately identifies all significant components; however, small proportions are frequently misassigned. When similar individuals are analyzed as a group, it is possible to estimate even the small proportions reliably.

### Phasing

The phased data were obtained for the 1000 Genomes Project from the open repository https://ftp.1000genomes.ebi.ac.uk/vol1/ftp/. *Sudmant et al. (2015)* used short-read Illumina DNA sequencing data to infer haplotype blocks in 26 human populations.

### Two modes

We have developed two modes: "diploid" and "haploid". For the "diploid" mode, we used the samples without pre-processing. For the "haploid" mode, we used the phasing information provided by 1000 Genomes and assumed that the parents were homozygous and simulated two homozygous "parents" for each sample. The admixture vector, used as

an informative prior for "diploid" and "haploid" modes, is generated by applying the Admixture tool (*Alexander, Novembre & Lange, 2009*) to the original "diploid" sample in supervised mode.

## Local ancestry bayesian approach (*PyLAE*)

**Input:**

**Tested Individual:** VCF file and (optional) admixture vector obtained by Admixture in supervised mode, as described above.

**Reference:** multi VCF file with putative ancestral populations corresponding to K component admixture.

$N$ is the number of positions overlapping between the reference and the individual.

The number of genomic positions should be significantly large to ensure dense SNP coverage, while not being too large since the markers should be uncorrelated. Therefore, we recommend removing positions based on high levels of pairwise linkage disequilibrium (LD-pruning). It is a common practice to exclude positions with pairwise genotypic $r^2 > 0.8$ within sliding windows of 50 SNPs (*Alexander, Novembre & Lange, 2009*).

### Stage 1. Bayesian posterior probability

Posterior probability $P(population|genotype)$ is calculated using the Bayes formula:

$$P(Population|genotype) = \frac{P(Population|genotype)}{P(genotype)} = \frac{P(Population \cap genotype)}{P(genotype)}$$
$$= \frac{P(Population)\ P(genotype|Population)}{P(genotype)}$$
$$= \frac{P(Population)\ P(genotype|Population)}{\sum_{k=1}^{K} P(genotype|Population_k)P(Population_k)}.$$

The prior probability of a population $P(population)$ is equal to the analyzed individual's admixture vector. We assume that adjacent positions have the same origin and the origin of all positions within a window of length **$L$**. Since the distance between two consecutive ancestry informative markers is sufficiently large, positions within the same window can be considered independent.

$$P(\{genotype_1\ldots genotype_L\}|\ population) = \prod_{i=1}^{L} P(genotype_i|population)$$

Therefore

$$P(population|genotype)\ = \frac{P(population)}{P(genotype)} \cdot \prod_{i=1}^{L} P(genotype_i|\ population)$$

$P(genotype_i|population)$ is estimated from observations, using genotypes of putative ancestral populations in position $i$

$$P(genotype) = \sum_{k=1}^{K} P(genotype|Population_k)P(Population_k)$$

Therefore, we have K posterior probabilities for every genotype, and the matrix of posterior probabilities $T \times K$, where $T = \frac{N}{L}$.

We are interested in determining the optimal sequence of the source populations along the genome. This problem is solved using the Viterbi algorithm (*Viterbi, 1967*). For computational efficiency, all calculations are performed in the log-space.

**Stage 2. Viterbi algorithm**

Our state-space $S$ consists of $K$ ancestral populations. Population labels ($i = 1..K$) are the hidden states in our model. The initial probabilities $\pi_i$ of being in the $i$th hidden state can either be assumed to be uniform with $1/K$ probability of each population or set to be equal to global admixture components. Transition probabilities $\alpha_{i,j}$ of transitioning from the state $i$ to $j$ are inversely proportional to the TreeMix distances between corresponding ancestral populations. Transition probability from state $i$ to $j$ $a_{ij}$ can be calculated from distances between ancestral populations; alternatively, the same value can be used for all pairs of populations. The emission probabilities $b_j(O)$ is calculated by the Bayes formula (above). At the first step of the algorithm $\delta_j(1) = \pi_j b_j(O_1)$. At each next step $\delta_j(t+1) = max_i \, \delta_i(t) \, a_{ij}(O_{t+1})$. The pointers $D_j(t+1)$ are stored for tracing back. $D_j(t+1) = argmax_i \, \delta_i(t) \, a_{ij}(O_{t+1})$. At the terminal state: $S_T = argmax_i \delta_i(T)$, $S_t = D_{S_{t+1}}(t+1)$.

For efficiency, the computation is performed in the log-space. The algorithm is implemented as a python script and can be adapted to analyze any organism using user-provided ancestral components, prior probabilities, and transition matrix.

### *Application of local ancestry*

Most of the currently living individuals are admixed to various degrees. To compare genotypes between populations and analyze selection signals, it is essential to identify and exclude introgressed regions containing non-representative genotypes. We have determined local ancestry profiles for all 2,504 analyzed individuals from the 1000 Genomes project. With our approach, even the admixed individuals that are typically excluded from the analysis (therefore, reducing the statistical power of the study) can be retained after masking the introgressed regions.

We have used the approach of *Chekalin et al. (2019)*, *Benítez-Burraco et al. (2021)* and annotated the VCFs with the ANNOVAR tool (*Wang, Li & Hakonarson, 2010*; *Yang & Wang, 2015*) using the hg19 human genome annotation and the refGene database (http://varianttools.sourceforge.net/Annotation/RefGene). The SNPs were classified by their location and function: 58% intergenic, 38% intronic, 5% ncRNA, and 1% exonic (64% of them are synonymous, and 36% are nonsynonymous). Next, we counted the synonymous and nonsynonymous SNPs per each KEGG pathway (*Ogata et al., 1999*; *Kanehisa, 2019*; *Kanehisa et al., 2019*).

It is reasonable to assume that pathways accumulate nonsynonymous SNPs at the same rate during neutral evolution; therefore, the pathways' enrichment scores can be approximated by a normal distribution. The pathways under selection will appear as outliers.

To analyze differences in numbers of SNPs per pathway between the two groups of populations (groups A and B), we need to calculate: (1) DSSE scores (synonymous) and

(2) DNSE scores (nonsynonymous) (*Chekalin et al., 2019*). Therefore, we perform the following steps:

(1) Find the number of (non)synonymous SNPs in groups A and B.

(a) Let $I$ represent the total number of studied pathways, and $i = 1,..., I$, be the number of the (non) synonymous SNP per i$^{th}$ pathway are nS(i) and nA(i). The expected fraction of (non) synonymous SNPs in individuals from group A is given by $p = \frac{nA}{nB+nA}$, where $nA$ is the amount of (non) synonymous SNPs in all KEGG pathways found in group A, $nB$ is the amount of (non) synonymous SNPs in all KEGG pathways found in group B. The fraction p$_i$ of A (non)synonymous SNPs in the i$^{th}$ KEGG pathway is $p_i = \frac{nA(i)}{nB(i)+nA(i)}$.

(2) The enrichment D(N)SE scores are computed for every pathway with continuity correction:

(a) $D(N/S)SE\ Score = \dfrac{(p - p_i) \pm \frac{1}{2(nS(i)+nA(i))}}{\sqrt{\frac{p(1-p)}{(nS(i)+nA(i))}}}$

(3) *P*-values are calculated using Bonferroni and Benjamini–Hochberg corrections and used to identify differentially enriched pathways. A pathway is considered to be differentially enriched if the adjusted *P*-value < 0.005 (*Benjamin et al., 2018*).

(4) To consider the excess of synonymous SNPs over nonsynonymous SNPs, we calculate enrichment scores for synonymous SNPs, DSSE. To be considered significant, the *P*-value of the nonsynonymous test is required to be below the corresponding *P*-value of the synonymous test for each pathway.

## Using *PyLAE* with different genomes and/or sets of markers

A different set of putative ancestral populations or a different set of markers require additional processing. First, we need to collect a database of putatively un-admixed individuals. If there is an existing validated set of ancestry informative features, these markers should run the admixture in supervised mode. For each self-reported ancestry, samples should be clustered based on their admixture profiles to identify subgroups within each self-reported ancestry. These subgroups are then examined using information about the studied population's history, and the most representative subset is retained. Then, putative ancestral populations (from 15 to 20 individuals per group) are generated for every ancestry. The validity and stability of the ancestral populations are evaluated using (1) PCA, (2) leave-one-out supervised admixture, and (3) by application of supervised admixture to the original dataset.

## RESULTS

### Investigation of the reference dataset

An intrinsic difficulty for benchmarking an ancestry prediction pipeline is the lack of the gold standard. We used the 1000 Genomes dataset for the benchmark and evaluated the quality of the dataset. At first, we have investigated the 1000 Genomes dataset to

determine intrinsic limitations to accuracy. Our first step is a comparison between phased and unphased data. *PyLAE* can process both phased and unphased data, which is both a blessing and a curse. It is a blessing since some genomes may not have phased data, and it is a curse since the accuracy may suffer. Therefore, we first evaluate the constraints that are imposed by different components of the pipeline. The first component is the calculation of the Admixture (*Alexander, Novembre & Lange, 2009*) profiles.

### Admixture profiles in Diploid vs. Haploid modes

Nine-dimensional admixture profiles were calculated for 2,504 individuals (130 K markers each) using supervised admixture. Every sample was analyzed first in diploid and then in haploid mode (Fig. 1).

Diploid and haploid admixture profiles were clustered within each reported ethnic origin (Fig. 2, Tables 1 and 2). African Ancestry SW, African Caribbean, Columbian, Gambian Mandinka, Mexican Ancestry, Peruvian, and Puerto-Rican showed 2–3 subgroups within each reported ethnicity, consistent with the complex population history of these populations.

We have investigated the difference between inferred haploid parental admixture profiles and diploid admixture profiles of analyzed individuals. We have computed an average of parental haploid admixture vectors for every individual and calculated the Euclidean distance between the vectors. The average distance for all individuals was 0.092. On average, the largest differences were observed for Toscani, Iberian, Columbian, Mexican Ancestry, and Puerto Rican individuals. In diploid mode, the average value of the Mediterranean component in Toscani individuals was 60%, South West Asian −23%, and Northern European −15%. In haploid mode, the Mediterranean component was 82%, and South-West Asian −18%. The same situation happened with the Iberian samples. In the diploid mode, the average value of the Mediterranean component was 55%, South West Asian −19%, and Northern European −24%. In the double haploid mode, the Mediterranean component was 72%, South West Asian −16%, and Northern European −12%. Similar inflation in the Mediterranean component was accompanied by a reduction in the Northern European components.

To investigate the source of this difference, we have identified the most significant admixture component for every sample and calculated the difference between the average parental value of this component and its diploid value. The difference ranged from −0.0057 to 0.3063; the mean difference was 0.0684, indicating that haploid admixture tends to increase the value of the largest component. The magnitude of the increment grows approximately linearly for components <0.87 and then reduces linearly for components above 0.87. Worldwide populations show different trends (Fig. 3). For individuals of European and East Asian descent, the trend is positive linear, and the larger the value, the more it is inflated. For South Asian individuals, there is no clear trend. For African and American individuals, there are multiple linear trends. The slope of the line is defined by the relative values of the admixture components. For example, among Columbians, a larger slope corresponds to the individuals where the average ratio of
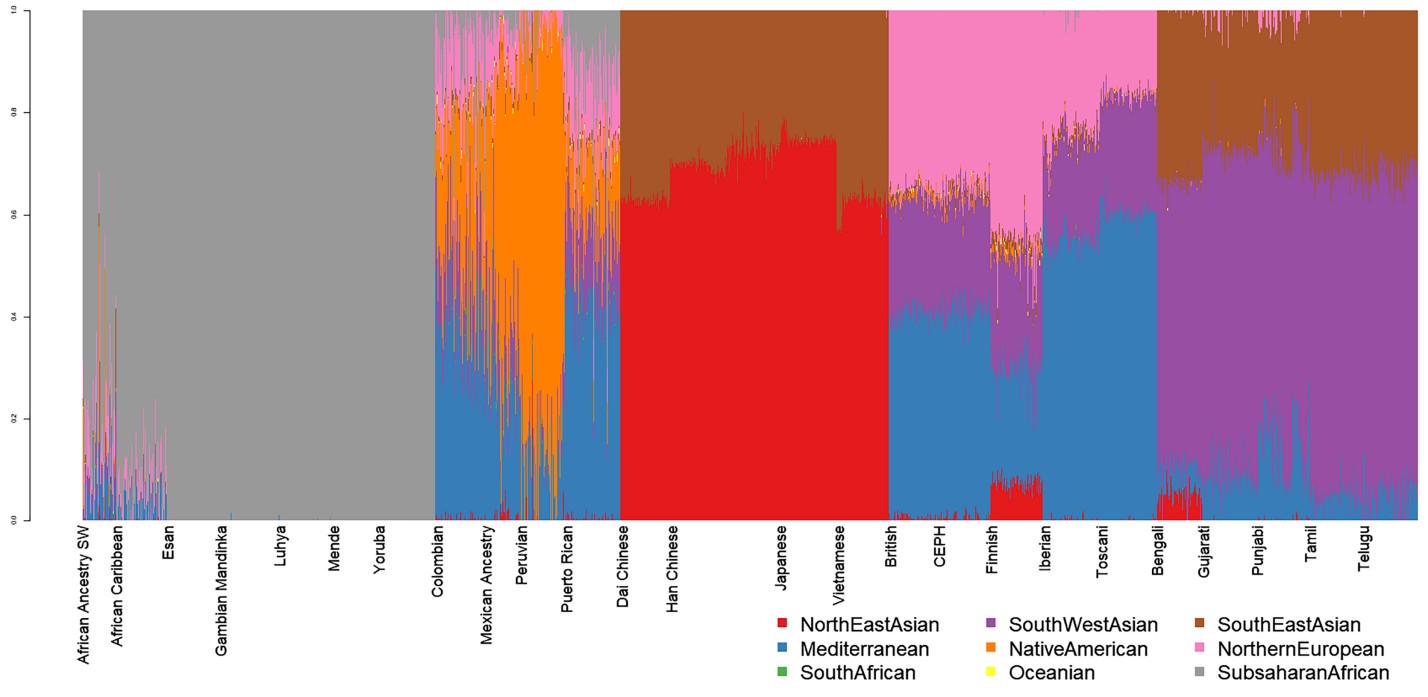

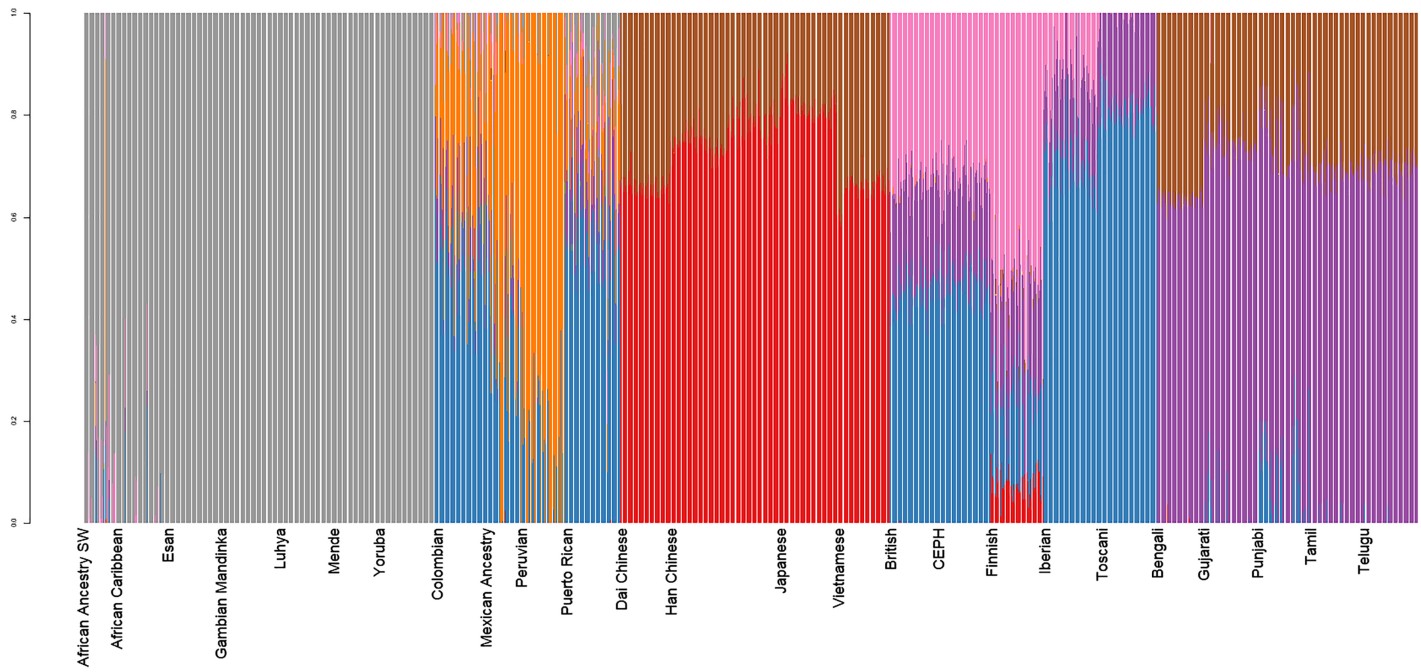

**Figure 1 Admixture plots for worldwide individuals from 1000 Genomes Project.** Top: diploid mode. Bottom: haploid mode. Colors: Red, North-East Asian; Orange, Native American; Green, South African; Purple, South-West Asian; Blue, Mediterranean; Yellow, Oceanian; Brow. Southeast Asian; Pink, Northern European; Grey, Sub-Saharan African.

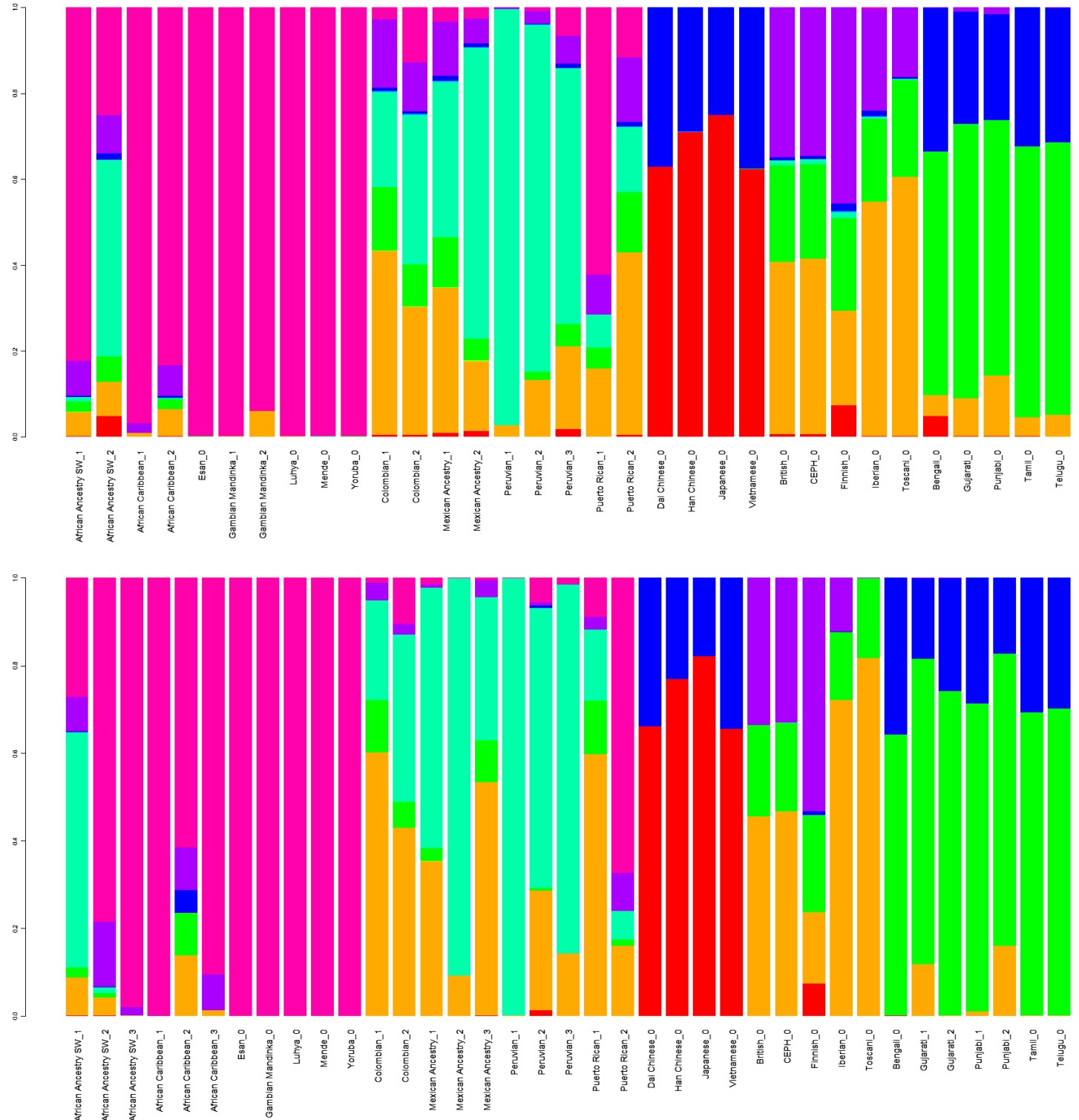

**Figure 2 Clustered Admixture plot for worldwide individuals from 1000 Genomes Project.** Clustered Admixture for 1000 Genomes individuals. Top: diploid mode. Bottom: haploid mode. Colors: Red, North-East Asian; Orange, Mediterranean; Light green, South African; Dark green, South-West Asian; Light blue, Native American; Blue, Oceanian; Dark blue, Southeast Asian; Purple, Northern European; Magenta, Sub-Saharan African.

**Table 1 Clusters of admixture vector clusters for 1000 Genomes individuals.**

| Super population name | Cluster | NorthEast Asian | Mediterranean | South African | SouthWest Asian | Native American | Oceanian | SouthEast Asian | Northern European | Subsaharan African | #ind |
|---|---|---|---|---|---|---|---|---|---|---|---|
| African | African Ancestry SW_1 | 5% | 8% | 0% | 6% | 46% | 0% | 1% | 9% | 25% | 3 |
| | African Ancestry SW_2 | 0% | 6% | 0% | 2% | 1% | 0% | 0% | 8% | 82% | 58 |
| | African Caribbean_1 | 0% | 6% | 0% | 2% | 0% | 0% | 1% | 7% | 83% | 35 |
| | African Caribbean_2 | 0% | 1% | 0% | 0% | 0% | 0% | 0% | 2% | 97% | 61 |
| | Esan_0 | 0% | 0% | 0% | 0% | 0% | 0% | 0% | 0% | 100% | 99 |
| | Gambian Mandinka_1 | 0% | 0% | 0% | 0% | 0% | 0% | 0% | 0% | 100% | 112 |
| | Gambian Mandinka_2 | 0% | 6% | 0% | 0% | 0% | 0% | 0% | 0% | 94% | 1 |
| | Luhya_0 | 0% | 0% | 0% | 0% | 0% | 0% | 0% | 0% | 100% | 99 |
| | Mende_0 | 0% | 0% | 0% | 0% | 0% | 0% | 0% | 0% | 100% | 85 |
| | Yoruba_0 | 0% | 0% | 0% | 0% | 0% | 0% | 0% | 0% | 100% | 108 |
| American Ancestry | Colombian_1 | 0% | 43% | 0% | 15% | 22% | 0% | 1% | 16% | 3% | 50 |
| | Colombian_2 | 0% | 30% | 0% | 10% | 35% | 0% | 1% | 11% | 13% | 40 |
| | Mexican Ancestry_1 | 1% | 16% | 0% | 5% | 68% | 0% | 1% | 6% | 3% | 27 |
| | Mexican Ancestry_2 | 1% | 34% | 0% | 12% | 36% | 0% | 1% | 13% | 3% | 37 |
| | Peruvian_1 | 0% | 3% | 0% | 0% | 97% | 0% | 0% | 0% | 0% | 25 |
| | Peruvian_2 | 2% | 19% | 0% | 5% | 60% | 0% | 1% | 7% | 7% | 27 |
| | Peruvian_3 | 0% | 13% | 0% | 2% | 81% | 0% | 0% | 3% | 1% | 33 |
| | Puerto Rican_1 | 0% | 42% | 0% | 14% | 15% | 0% | 1% | 15% | 12% | 90 |
| | Puerto Rican_2 | 0% | 16% | 0% | 5% | 8% | 0% | 0% | 9% | 62% | 3 |
| East Asian | Dai Chinese_0 | 63% | 0% | 0% | 0% | 0% | 0% | 37% | 0% | 0% | 92 |
| | Han Chinese_0 | 71% | 0% | 0% | 0% | 0% | 0% | 29% | 0% | 0% | 197 |
| | Japanese_0 | 75% | 0% | 0% | 0% | 0% | 0% | 25% | 0% | 0% | 104 |
| | Vietnamese_0 | 62% | 0% | 0% | 0% | 0% | 0% | 38% | 0% | 0% | 99 |
| European | British_0 | 1% | 40% | 0% | 22% | 1% | 0% | 1% | 35% | 0% | 91 |
| | CEPH_0 | 1% | 41% | 0% | 22% | 1% | 0% | 1% | 35% | 0% | 99 |
| | Finnish_0 | 7% | 22% | 0% | 22% | 1% | 0% | 2% | 46% | 0% | 93 |
| | Iberian_0 | 0% | 55% | 0% | 19% | 0% | 0% | 1% | 24% | 0% | 107 |
| | Toscani_0 | 0% | 60% | 0% | 23% | 0% | 0% | 0% | 16% | 0% | 107 |
| South Asian | Bengali_0 | 5% | 5% | 0% | 57% | 0% | 0% | 33% | 0% | 0% | 86 |
| | Gujarati_0 | 0% | 9% | 0% | 64% | 0% | 0% | 26% | 1% | 0% | 103 |
| | Punjabi_0 | 0% | 14% | 0% | 60% | 0% | 0% | 25% | 2% | 0% | 96 |
| | Tamil_0 | 0% | 4% | 0% | 63% | 0% | 0% | 32% | 0% | 0% | 102 |
| | Telugu_0 | 0% | 5% | 0% | 64% | 0% | 0% | 31% | 0% | 0% | 102 |

**Note:**
Admixture vectors were obtained using admixture in supervised mode.

**Table 2 Haploid admixture vector clusters for 1000 Genomes individuals.**

| Super population name | Cluster | NorthEast Asian | Mediterranean | South African | SouthWest Asian | Native American | Oceanian | SouthEast Asian | Northern European | Subsaharan African | #parents |
|---|---|---|---|---|---|---|---|---|---|---|---|
| African | African Ancestry SW_1 | 0% | 0% | 0% | 0% | 0% | 0% | 0% | 2% | 98% | 71 |
| | African Ancestry SW_2 | 0% | 4% | 0% | 1% | 1% | 0% | 0% | 15% | 78% | 45 |
| | African Ancestry SW_3 | 0% | 9% | 0% | 2% | 54% | 0% | 0% | 8% | 27% | 6 |
| | African Caribbean_1 | 0% | 14% | 0% | 10% | 0% | 0% | 5% | 10% | 62% | 6 |
| | African Caribbean_2 | 0% | 1% | 0% | 0% | 0% | 0% | 0% | 8% | 90% | 26 |
| | African Caribbean_3 | 0% | 0% | 0% | 0% | 0% | 0% | 0% | 0% | 100% | 160 |
| | Esan_0 | 0% | 0% | 0% | 0% | 0% | 0% | 0% | 0% | 100% | 198 |
| | Gambian Mandinka_0 | 0% | 0% | 0% | 0% | 0% | 0% | 0% | 0% | 100% | 226 |
| | Luhya_0 | 0% | 0% | 0% | 0% | 0% | 0% | 0% | 0% | 100% | 198 |
| | Mende_0 | 0% | 0% | 0% | 0% | 0% | 0% | 0% | 0% | 100% | 170 |
| | Yoruba_0 | 0% | 0% | 0% | 0% | 0% | 0% | 0% | 0% | 100% | 216 |
| American | Colombian_1 | 0% | 43% | 0% | 6% | 38% | 0% | 0% | 2% | 11% | 86 |
| | Colombian_2 | 0% | 60% | 0% | 12% | 22% | 0% | 0% | 4% | 1% | 94 |
| | Mexican Ancestry_1 | 0% | 35% | 0% | 3% | 59% | 0% | 0% | 1% | 2% | 60 |
| | Mexican Ancestry_2 | 0% | 53% | 0% | 10% | 33% | 0% | 0% | 4% | 1% | 46 |
| | Mexican Ancestry_3 | 0% | 9% | 0% | 0% | 91% | 0% | 0% | 0% | 0% | 22 |
| | Peruvian_1 | 1% | 27% | 0% | 1% | 64% | 0% | 1% | 1% | 6% | 48 |
| | Peruvian_2 | 0% | 14% | 0% | 0% | 84% | 0% | 0% | 0% | 1% | 50 |
| | Peruvian_3 | 0% | 0% | 0% | 0% | 100% | 0% | 0% | 0% | 0% | 72 |
| | Puerto Rican_1 | 0% | 59% | 0% | 12% | 16% | 0% | 0% | 3% | 10% | 180 |
| | Puerto Rican_2 | 0% | 16% | 0% | 1% | 6% | 0% | 0% | 9% | 67% | 6 |
| East Asian | Dai Chinese_0 | 66% | 0% | 0% | 0% | 0% | 0% | 34% | 0% | 0% | 184 |
| | Han Chinese_0 | 77% | 0% | 0% | 0% | 0% | 0% | 23% | 0% | 0% | 394 |
| | Japanese_0 | 82% | 0% | 0% | 0% | 0% | 0% | 18% | 0% | 0% | 208 |
| | Vietnamese_0 | 65% | 0% | 0% | 0% | 0% | 0% | 35% | 0% | 0% | 198 |
| European | British_0 | 0% | 46% | 0% | 21% | 0% | 0% | 0% | 34% | 0% | 182 |
| | CEPH_0 | 0% | 47% | 0% | 20% | 0% | 0% | 0% | 33% | 0% | 198 |
| | Finnish_0 | 7% | 17% | 0% | 22% | 0% | 0% | 1% | 53% | 0% | 186 |
| | Iberian_0 | 0% | 72% | 0% | 16% | 0% | 0% | 0% | 12% | 0% | 214 |
| | Toscani_0 | 0% | 82% | 0% | 18% | 0% | 0% | 0% | 0% | 0% | 214 |
| South Asian | Bengali_0 | 0% | 0% | 0% | 64% | 0% | 0% | 36% | 0% | 0% | 172 |
| | Gujarati_1 | 0% | 0% | 0% | 74% | 0% | 0% | 26% | 0% | 0% | 180 |
| | Gujarati_2 | 0% | 12% | 0% | 70% | 0% | 0% | 18% | 0% | 0% | 26 |
| | Punjabi_1 | 0% | 16% | 0% | 67% | 0% | 0% | 17% | 0% | 0% | 89 |

(Continued)
| Super population name | Cluster | NorthEast Asian | Mediterranean | South African | SouthWest Asian | Native American | Oceanian | SouthEast Asian | Northern European | Subsaharan African | #parents |
|---|---|---|---|---|---|---|---|---|---|---|---|
| | Punjabi_2 | 0% | 1% | 0% | 70% | 0% | 0% | 29% | 0% | 0% | 103 |
| | Tamil_0 | 0% | 0% | 0% | 69% | 0% | 0% | 31% | 0% | 0% | 204 |
| | Telugu_0 | 0% | 0% | 0% | 70% | 0% | 0% | 30% | 0% | 0% | 204 |

**Note:**
Supervised admixture using phased haploid data.

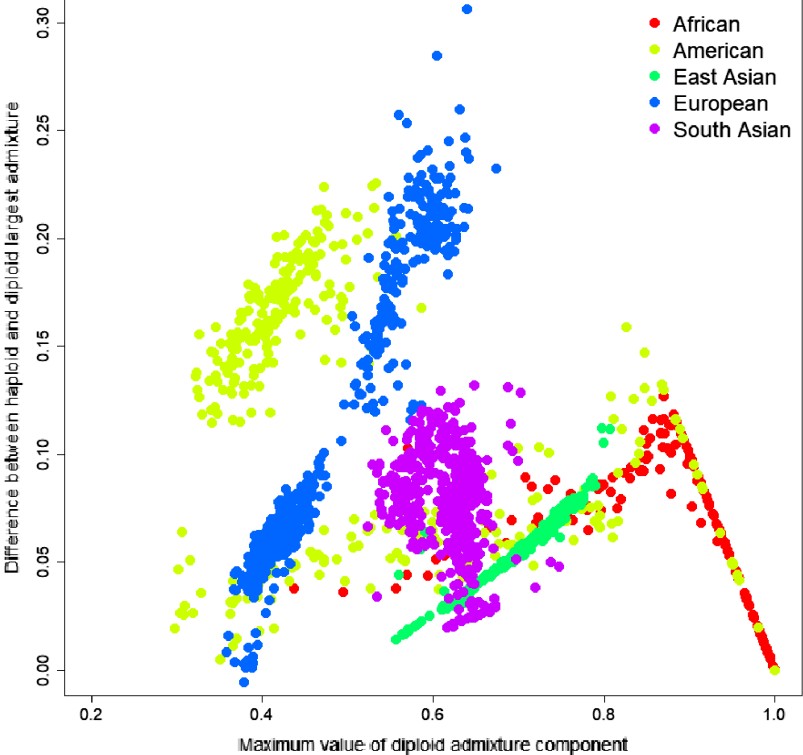

**Figure 3 Difference between the largest admixture component in the aggregated haploid and diploid cases.** We have identified the largest admixture component for every sample and calculated the difference between the average parental value of this component and its diploid value. The mean difference was 0.0684, indicating that haploid admixture tends to slightly increase the value of the largest component determined by the diploid mode.

Mediterranean to the sum of Sub-Saharan African and Native American components is 2.07. In comparison, a smaller slope corresponds to a 0.56 ratio.

### Population-specific accuracy limitations of the admixture-based approach

We have investigated the accuracy limitations of the dataset and admixture-based approaches. First, for each individual's admixture vector, we have calculated the nearest population, based on Euclidean distance between admixture vectors, using the leave-one-out approach; then we have computed fractions of self-hits (the nearest population agrees with the population label of the individual), and fractions of super population

**Table 3  GPS and reAdmix.**

| Super-population | Population | Number of individuals | GPS, fraction of populations self-hits | GPS, fraction of super-population hits | reAdmix, fraction of populations self-hits | reAdmix, fraction of super-population hits |
|---|---|---|---|---|---|---|
| African | African Ancestry SW | 61 | 0.34 | 0.85 | 0.58 | 0.98 |
| | African Caribbean | 96 | 0.56 | 0.97 | 0.47 | 0.99 |
| | Esan | 99 | 0.07 | 1.00 | 0.96 | 1.00 |
| | Gambian Mandinka | 113 | 0.01 | 1.00 | 0.01 | 1.00 |
| | Luhya | 99 | 0.07 | 1.00 | 0.00 | 1.00 |
| | Mende | 85 | 0.92 | 1.00 | 0.01 | 1.00 |
| | Yoruba | 108 | 0.19 | 1.00 | 0.10 | 1.00 |
| American | Colombian | 94 | 0.65 | 0.96 | 0.46 | 0.70 |
| | Mexican Ancestry | 64 | 0.5 | 0.97 | 0.17 | 0.89 |
| | Peruvian | 85 | 0.8 | 0.98 | 0.86 | 0.96 |
| | Puerto Rican | 104 | 0.85 | 0.93 | 0.63 | 0.78 |
| East Asian | Dai Chinese | 93 | 0.62 | 1.00 | 0.62 | 1.00 |
| | Han Chinese | 208 | 0.71 | 1.00 | 0.25 | 1.00 |
| | Japanese | 104 | 0.96 | 1.00 | 0.89 | 1.00 |
| | Vietnamese | 99 | 0.41 | 1.00 | 0.37 | 1.00 |
| European | British | 91 | 0.58 | 1.00 | 0.89 | 0.99 |
| | CEPH | 99 | 0.57 | 1.00 | 0.13 | 0.99 |
| | Finnish | 99 | 0.97 | 1.00 | 0.90 | 0.99 |
| | Iberian | 107 | 0.94 | 1.00 | 0.84 | 0.98 |
| | Toscani | 107 | 1 | 1.00 | 0.99 | 0.99 |
| Southeast Asian | Bengali | 86 | 0.9 | 1.00 | 0.72 | 1.00 |
| | Gujarati | 103 | 0.69 | 1.00 | 0.52 | 0.99 |
| | Punjabi | 96 | 0.59 | 1.00 | 0.46 | 0.98 |
| | Tamil | 102 | 0.59 | 1.00 | 0.73 | 1.00 |
| | Telugu | 102 | 0.34 | 1.00 | 0.34 | 1.00 |

**Note:**
Results of GPS and reAdmix analyses. For each population, we report factions of self-hits (when the tool correctly identifies the population label) and superpopulation hits (when a superpopulation is correctlyidentified).

self-hits (the agreement is at the level of super-populations). Admixed populations like African American, African Caribbean, Colombian, Mexican in the USA, Peruvian, and Puerto Rican have the lowest assignment accuracy at both population and subpopulation levels. We also noticed that West African populations, such as Yoruba, Esan, Gambian Mandinka, and Mende, are frequently misclassified. Although they speak languages from the same Niger-Congo group, their genomes are not identical (*Skoglund et al., 2017*; *Fan et al., 2019*). Simplified representation using nine admixture components is artificially "collapsing" those groups to a single point in 9-dimensional space.

GPS and reAdmix (Table 3, Fig. 4) analyses suggest that population labels were not 100% accurate but were correct at the superpopulation level. For example, about 30% of Yoruba individuals were mapped to Yoruba reference, but all were mapped to various African reference populations. Several factors can explain this: first, there is true
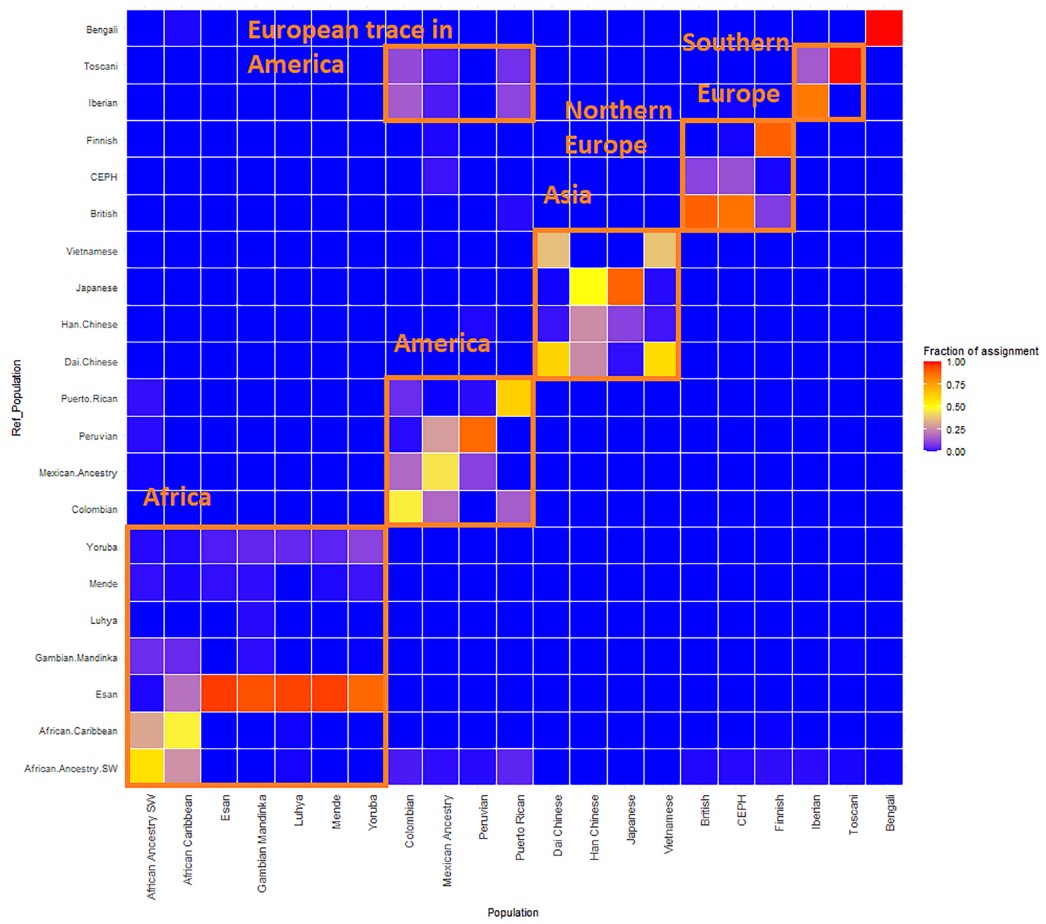

**Figure 4 ReAdmix assignment matrix heatmap.** reAdmix analysis shows that population labels may not be 100% accurate; however, they are correct at the superpopulation level. This may be explained by the presence of admixed individuals in the dataset, and by shared histories by various groups. The assignment matrix indicates the percentage assignment. The color intensity of the heatmap shows the probability of an individual from a reference population (columns) being assigned to a given population (rows). The darkest shade indicates the highest probability and the lightest shade indicates the lowest.

heterogeneity of individuals and fuzzy boundaries between populations; second, there may be a human error factor on behalf of the sampled individual or a researcher collecting data; the third factor is the unreported admixture from different populations. Native and African American individuals appear to be the most affected by these factors; this observation agrees with known historical events. Therefore, we expect to see admixed regions along the genomes of affected individuals.

## Performance of the *PyLAE* algorithm

We have tested the *PyLAE* algorithm on 2,504 individuals of the 1000 Genomes dataset using: diploid and haploid versions, with informative and noninformative priors. Informative prior is assumed to be equal to the global admixture vector. The noninformative prior is uniform, assuming equal probabilities for a region to be from any of the K reference populations.

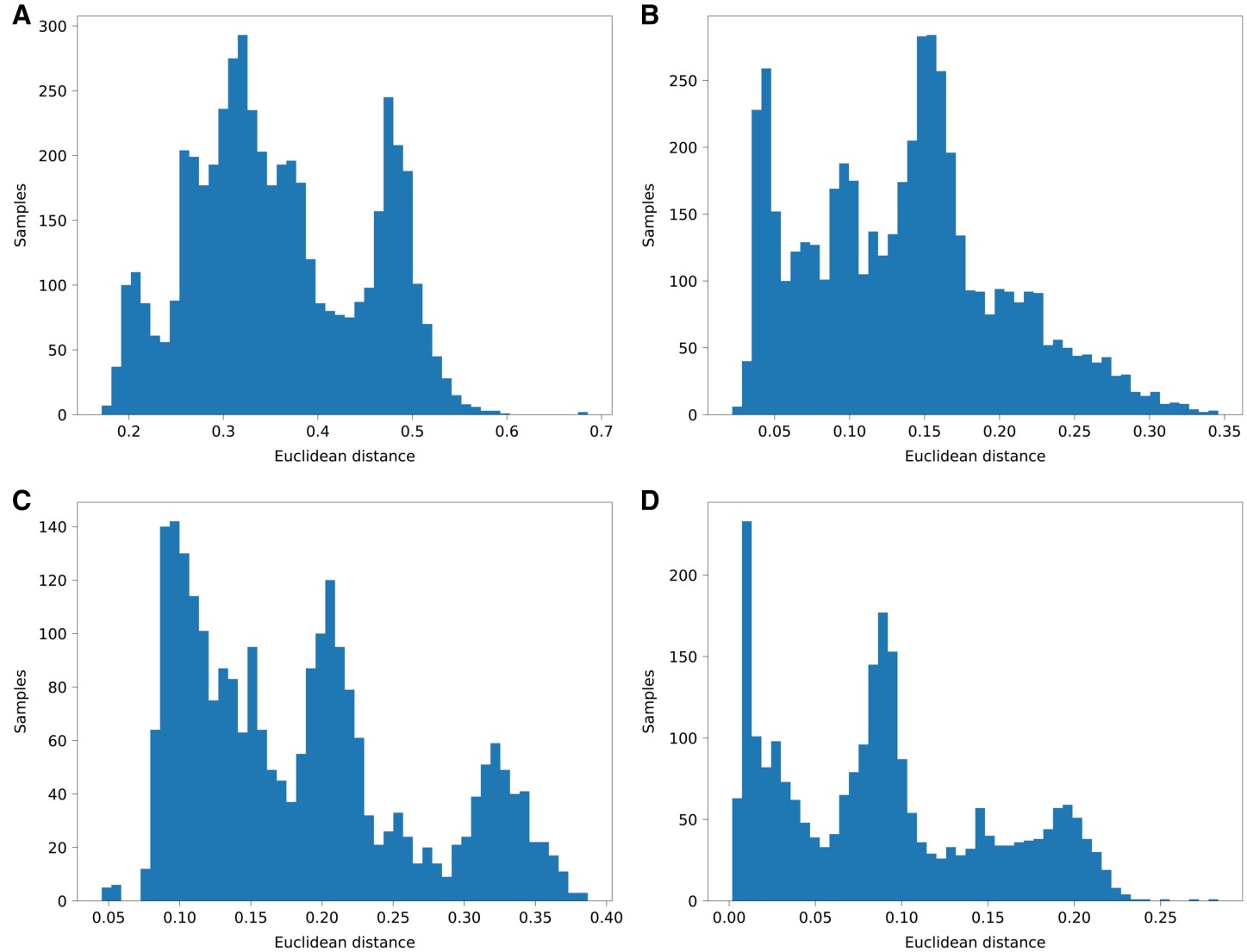

**Figure 5 Histograms of euclidean distances of aggregated PyLAE results *vs.* admixture vectors.** Euclidean distances of aggregated PyLAE results *vs.* admixture vectors for 2,504 from 1000 Genomes. (A) 'Haploid' mode, no admixture prior. (B) 'Haploid' mode, with admixture prior. (C) "Diploid" mode, no admixture prior. (D) "Diploid" mode, with admixture prior.

Since we do not know the origins of regions of the genome, we chose the agreement with the global admixture vector as a measure of performance. For each analyzed individual, we have computed a total fraction of a genome attributed to each of the K components and compared it to the global admixture vector. We have calculated Euclidean distance and Pearson correlation of aggregated results and admixture vectors. The results are shown in Fig. 5 and Table 4.

We compared *PyLAE* with the current local ancestry gold-standard tool RFMix (*Maples et al., 2013*; *Uren, Hoal & Möller, 2020*). We used the 1000 Genomes dataset and have selected 150K ancestry-informative markers. *PyLAE* and RFMix use different inputs and require separate data preparation steps. For RFMix, the following actions were performed: (1) phase ancestral samples (2) obtain genetic maps. The phasing was performed with

**Table 4 Average Euclidean distance between global admixture vectors and aggregated PyLAE results in diploid mode.**

| Population | Average Euclidean distance between the global admixture vector and aggregated PyLAE results in diploid mode | |
| --- | --- | --- |
| | Informative prior | Uniform prior |
| Yoruba | 0.000747008 | 0.013042043 |
| Esan | 0.000770357 | 0.013533177 |
| Mende | 0.000856896 | 0.018231929 |
| Iberian | 0.001331954 | 0.012167398 |
| Gambian Mandinka | 0.001365033 | 0.021373737 |
| Toscani | 0.001539027 | 0.019696897 |
| Luhya | 0.001826951 | 0.027512649 |
| African Caribbean | 0.004803319 | 0.0301569 |
| Puerto Rican | 0.005675916 | 0.017629195 |
| CEPH | 0.006072862 | 0.011648137 |
| Japanese | 0.006491083 | 0.046715682 |
| British | 0.006925157 | 0.012265145 |
| Han Chinese | 0.007304674 | 0.045295263 |
| Kinh | 0.007519174 | 0.04711133 |
| Southern Han Chinese | 0.0075584 | 0.044229833 |
| Dai Chinese | 0.008293571 | 0.051361167 |
| Kinh Vietnamese | 0.008600211 | 0.05305063 |
| African Ancestry SW | 0.010087603 | 0.03448744 |
| Colombian | 0.01190426 | 0.030931471 |
| Peruvian | 0.016376169 | 0.069468746 |
| Mexican Ancestry | 0.018915794 | 0.049218005 |
| Finnish | 0.02410449 | 0.038832151 |
| Punjabi | 0.026287313 | 0.08384595 |
| Gujarati | 0.028881709 | 0.093738688 |
| Telugu | 0.033376937 | 0.115691247 |
| Tamil | 0.036767906 | 0.120281378 |
| Bengali | 0.04353631 | 0.110281922 |

**Note:**
Accuracy of PyLAE is assessed using the difference between aggregated results predicted by PyLAE and independently calculated admixture vector.

Beagle (*Browning, Zhou & Browning, 2018*). *PyLAE* window size was set to 15, 30, 50, and 100 SNPs; no genetic map information was used. *PyLAE* was tested with the prior (results of admixture in supervised mode) and without prior, only the 'phased' case. We have also tried various values of the transition penalty. The probability of changing the population assignment (transition probability) was modeled as $a_{i \neq j} = \frac{1}{n+x}$, where $n$ is the number of populations and $x$ is the penalty, that was ranged from 0 (uniform prior), to 10,000.

We aggregated the results according to the following rules: for *PyLAE*–we have computed a fraction of windows assigned to each ancestral population; for RFMix, we have calculated a fraction of markers assigned to each ancestral population. Then the results

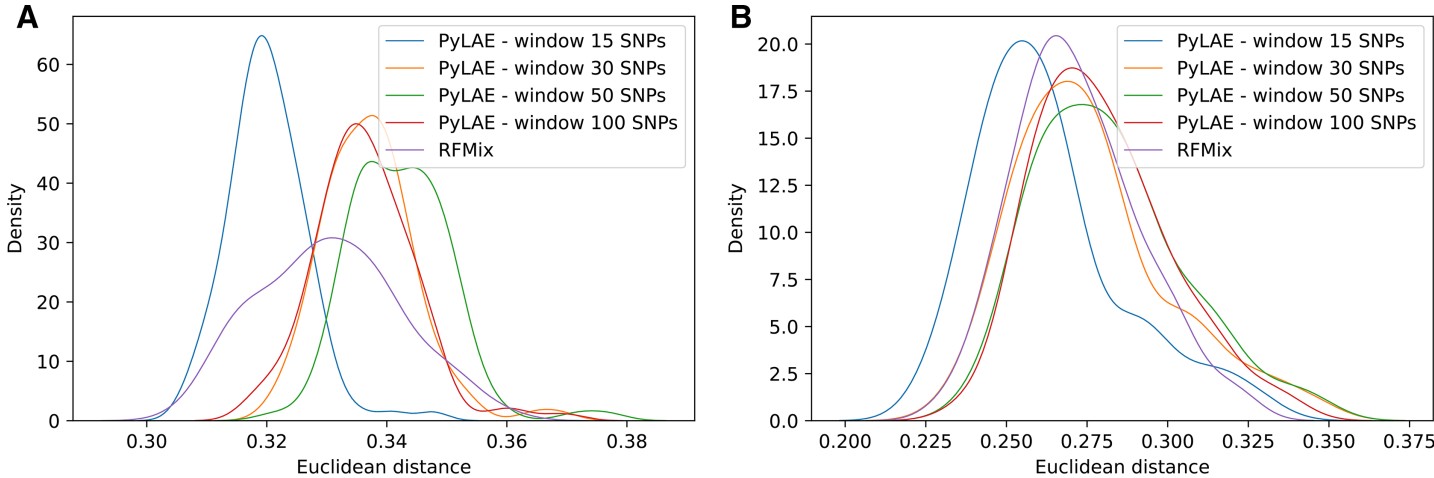

**Figure 6 Comparison of performance of PyLAE and RFMix using 1000 Genomes database.** Four PyLAE runs were performed: 15, 30, 50, and 100 SNP windows. RFMix program was used with default parameters. Euclidean distance was computed between the admixture profiles and predicted aggregated ancestry for TSI (A) and CLM (B) samples.

**Table 5 Pearson correlation coefficients between aggregated local ancestry predictions by PyLAE and RFMix and global ancestry, represented by admixture vectors.** For individuals with East Asian and European ancestry, PyLAE has a higher correlation with corresponding global ancestral composition compared to RFMix, while for African, American, and South Asian individuals, RFMix had a higher correlation. It is important to notice that global and aggregated local ancestries were in agreement for both programs; the lowest correlation coefficient was 0.8676.

| Superpopulation | PyLAE | RFMix |
|---|---|---|
| AFR | 0.9893 | 0.9996 |
| AMR | 0.9675 | 0.9851 |
| EAS | 0.9783 | 0.9684 |
| EUR | 0.9744 | 0.9699 |
| SAS | 0.9185 | 0.9595 |

were compared with the global ancestry assignments calculated by Admixture (*Alexander, Novembre & Lange, 2009*).

We simulated admixed individuals with the known origin of each genomic fragment and computed the fraction of correctly assigned positions. The choice of the optimal window size (measured in the number of SNPs per window) is determined by the ancestral composition of the analyzed sample (Fig. 6). Overall, an increase in the transition penalty suppresses switches between predicted populations and results in smoother prediction. Therefore, we recommend using the penalty of 10,000 or higher unless there are reasons to believe that the sample is highly mosaic.

Table 5 shows that for individuals with East Asian and European ancestry, *PyLAE* has a higher correlation with corresponding global ancestral composition compared to RFMix. For African, American, and South Asian individuals, RFMix had a higher correlation. It is important to notice that global and aggregated local ancestries were in
agreement for both programs; the lowest correlation coefficient was 0.8676. In the 1000 Genomes dataset, African, American, and South Asian samples are more admixed than East Asian and European samples.

*PyLAE* and RFMix have correctly determined the origin for 98.6% and 99% of genomic positions across all simulated samples, respectively. Predictions made by RFMix and *PyLAE* were highly concordant: the predictions agreed on 98.52% of the genome. Therefore, we propose that *PyLAE* and RFMix complement each other and be used as a part of the same local ancestry pipeline to improve prediction accuracy.

## Application of local ancestry

Using the ANNOVAR (*Wang, Li & Hakonarson, 2010*; *Yang & Wang, 2015*) and snpEFF (*Cingolani et al., 2012*), we have performed variation annotations and calculated the counts of synonymous and nonsynonymous SNPs. We have conducted this analysis in several modes: (1) comparing British, Finnish, and CEU samples with African American and African Caribbean samples; (2) comparing British, Finnish, and CEU samples with all African samples; (3) comparing British, Finnish, and CEU samples with African samples from Africa, selecting Northern European component in European samples and Sub-Saharan African component in African samples; (4) comparing British, Finnish and CEU samples with African American and African Caribbean samples, selecting Northern European component in European samples and the Sub-Saharan African component in African samples; (5) comparing British, Finnish and CEU samples with African samples from Africa, selecting the Sub-Saharan African component in African samples; (6) comparing British, Finnish and CEU samples with African Americans and the African Caribbean, selecting the Sub-Saharan African component in African samples.

Since we use a nine-component representation, all Africans in our datasets were assigned to the same Sub-Saharan African component. At the same time, African Americans have various admixture levels from other components. Therefore, local ancestry mode only partitions genomes of European and African American individuals.

On average, *PyLAE* has higher enrichment scores compared to the whole-genome approach. Comparing the enrichment scores between differentially enriched pathways (*P*-value < 0.01), in 91 cases, *PyLAE* resulted in higher enrichment scores and 41-in lower scores for African Americans. If the significance cut-off is lowered, *PyLAE* results in a higher number of differentially enriched pathways (Fig. 7).

Application of *PyLAE* to African American samples allowed the detection of eleven differentially enriched pathways that were not detected when entire African Americans' genomes were used but appeared when Europeans were compared with Africans from Africa. According to *PyLAE*, the following pathways were enriched in nonsynonymous SNPs: **Africans:** Lipoic acid metabolism, Vitamin B6 metabolism, Fatty acid biosynthesis, Notch signaling pathway, Type I diabetes mellitus, Allograft rejection, Cell adhesion molecules (CAMs), Phagosome, Malaria, Viral myocarditis, HTLV-I infection; **Europeans:** Nitrogen metabolism, Mismatch repair, Cell cycle, Glyoxylate and dicarboxylate metabolism, Porphyrin and chlorophyll metabolism, Insulin resistance,

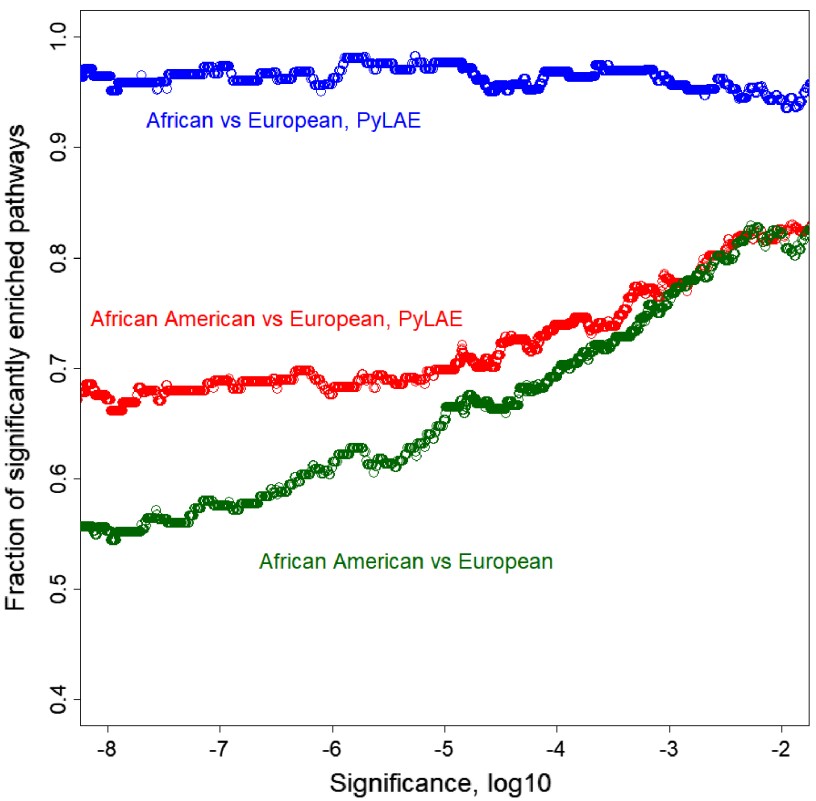

**Figure 7 Fraction of significantly enriched pathways as a function of a significance cut-off.** PyLAE has higher enrichment scores compared to the whole-genome approach. Comparing the enrichment scores between differentially enriched pathways (*P*-value < 0.01), in 91 cases, PyLAE resulted in higher enrichment scores and 41 in lower scores for African Americans. If the significance cut-off is lowered, PyLAE results in a higher number of differentially enriched pathways.

Tight junction, Selenocompound metabolism, Oocyte meiosis, Amyotrophic lateral sclerosis (ALS), Protein digestion and absorption.

Dependence on ethnicity efficiency has been reported in processes and conditions related to the above-listed pathways: vitamin B6 metabolism (*Gong et al., 2014*), fatty acid biosynthesis (*Sergeant et al., 2012*; *Ameur et al., 2012*; *Kothapalli, Park & Brenna, 2020*), resistance to malaria (*Singh & Dhar, 2014*; *Gelabert et al., 2017*), the prevalence of viral myocarditis (*Shaboodien et al., 2013*) and amyotrophic lateral sclerosis (*Rechtman et al., 2015*; *Henning et al., 2021*). According to recent studies (*Nédélec et al., 2016*; *Singh et al., 2020*), individuals with African ancestry have a stronger inflammatory response and reduced intracellular bacterial growth. Maintaining a strong immune response can have adverse side effects, such as autoimmune disease. Therefore, phagosome efficiency declined after migration out of Africa since Europeans were exposed to fewer pathogens, reducing immune response over generations.

In summary, *PyLAE* is an easy-to-use, accurate and fast tool for local ancestry analysis that can be used with minimal prior information about the genome or samples.

## Availability and requirements

Project name: Python Local Admixture Estimation (*PyLAE*)

Project home page: https://github.com/smetam/PyLAE

Operating system(s): Platform independent, Linux is required for pre-processing

Programming language: Python, Shell

Other requirements: Python 3.5 and higher; bcftools (pre-processing)

License: Creative Commons.

# LIST OF ABBREVIATIONS

**AIM**  Ancestry Informative Markers

**GWAS**  Genome-Wide Association Study

**SNP**  Single Nucleotide Polymorphism

## Funding

The authors received no funding for this work.

## Competing Interests

Nikita Moshkov is a part-time bioinformatician for Atlas Biomed.

Tatiana Tatarinova is an academic editor for PeerJ.

## Author Contributions

- Nikita Moshkov analyzed the data, prepared figures and/or tables, authored or reviewed drafts of the paper, and approved the final draft.
- Aleksandr Smetanin performed the experiments, analyzed the data, authored or reviewed drafts of the paper, and approved the final draft.
- Tatiana V. Tatarinova conceived and designed the experiments, analyzed the data, prepared figures and/or tables, authored or reviewed drafts of the paper, and approved the final draft.

## Data Availability

The data is available at GitHub: https://github.com/smetam/pylae

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
