# Peer review of "Local ancestry prediction with PyLAE"

_PeerJ, doi:10.7717/peerj.12502_

## Round 0.1 · original submission · Major Revisions

Dear Dr. Smetanin and colleagues:

Thanks for submitting your manuscript to PeerJ. I have now received two independent reviews of your work, and as you will see, the reviewers raised some concerns about the research. Despite this, these reviewers are optimistic about your work and the potential impact it will lend to research on identifying local ancestry in sequence datasets. Thus, I encourage you to revise your manuscript, accordingly, taking into account all of the concerns raised by the reviewers.

Both reviewers found your manuscript to be mostly well written and organized. However, each found some methodological limitations, and there is agreement that a few more analyses may help the overall study by showing PyLAE outperforms similar approaches.

In essence, your work goes back to the old idea of ancestry informative markers (i.e. just using SNP frequencies), which ignores much more information in the haplotypes. It is unfortunate that you don't compare the performance to RFMix or other haplotye-based methods. This would certainly strengthen your argument for the superiority of PyLAE.

It appears that some of the figures are confusing or missing information.

Please address all of these issue raised by both reviewers.

I look forward to seeing your revision, and thanks again for submitting your work to PeerJ.

Good luck with your revision,

-joe

Reviewer 1 ·

Basic reporting

In “Local ancestry prediction with PyLAE” Smetanin et al present a new method, PyLAE, for inferring local ancestry. Local ancestry programs are often complicated and PyLAE’s straightforward methodology (Bayesian posterior probabilities followed by Viterbi processing) makes it a potentially useful tool. The manuscript is generally well-written and easy to understand however I have some concerns about how well the algorithm is validated, and how other analyses presented in the manuscript serve to illustrate PyLAE’s utility. These concerns are outlined in more detail in “Validity of Findings” and “General Comments” below.
The authors point out that there is “always room to develop a user-friendly, fast, accurate” approaches for local ancestry (line 40), and I agree; very few programs are well-suited to every research question and new programs that are well-validated may fill a previously unoccupied niches, supplant existing programs, or improve accessibility of local ancestry methods to more researchers. With that in mind, I still believe that the authors should provide more motivation for developing a new local ancestry program or provide additional information in the Introduction or elsewhere on how PyLAE represents an improvement over other local ancestry inference programs, or what knowledge gaps may be filled with PyLAE. Inclusion of this information could provide a stronger ecntral purpose for the manuscript, as some of the manuscript’s different sections in Results seem to don’t seem to tie together well, even if the results presented within each section may be independently interesting if explored in more depth.
The figures and tables presented in the manuscript are relevant to the results discussed in the manuscript and appear to be of good quality, however I have concerns with the colors of Figures 1-3 and I have few suggestions to improve the interpretability of figure 4. These comments can be found in the “General Comments” below.

Experimental design

Although the algorithm for PyLAE is generally well-explained in the methods section, experiments to validate PyLAE are generally lacking. The only experiment that attempts to validate PyLAE’s accuracy is a correlation test between PyLAE’s local ancestry inference and global ancestry inference determined by ADMXITURE (presented in Figure 5). This experiment is a good start to validate PyLAE’s output, but it is insufficient on its own to validate the program. I make specific suggestions on PyLAE validation in “General Comments” below.

Validity of the findings

The manuscript lacks a single central finding though there are potentially interesting findings on the inflation of admixture proportions presented in the “Investigation of the reference dataset” (lines 208-289), and another potentially interesting finding about power to investigate selection on different molecular pathways with and without local ancestry information presented in “Application of Local Ancestry”. From the title and abstract, I would anticipate that most findings from the manuscript would be found in the “Performance of the PyLAE algorithm” section, but just one figure (Figure 5) is found in this section (specific suggestions to improve this section are recorded in “General Comments”). As such, the summary finding that PyLAE is “an easy-to-use and fast tool to perform ancestry decomposition” (lines 369-370) is not well-supported.
I note that the section “Application of Local Ancestry” provides a potentially useful illustration of an analysis which can be improved with local ancestry information and is an appropriate section for a manuscript that introduces a new program, but this section demonstrates the potential utility of PyLAE without addressing its accuracy. Since the authors had not previously compared PyLAE to existing local ancestry programs in depth or otherwise demonstrated its accuracy beyond simple correlation with global ancestry proportions, it is difficult to interpret the findings presented in this section.

Additional comments

Major Points:
1. As mentioned above, the manuscript seems to struggle tying the different analyses presented in the manuscript into one cohesive message. Each of the three different sections in Results presents findings that could be of potential interest to readers if explored in more detail but the authors should clarify more explicitly how the result is relevant to the central message of the manuscript.
2. I strongly suggest revision of the abstract to include the full scope of what is presented in the manuscript. As it is currently written, the abstract suggests that the manuscript’s focus will be nearly exclusively on the implementation of PyLAE. Instead, though PyLAE remains a substantial portion of the methods section, it is a minor topic in the results section. Additionally, certain properties of PyLAE mentioned in the abstract are not mentioned in the text (i.e., efficiency, speed, presentation of results, number of populations).
3. If the subject of the manuscript is to introduce PyLAE as a program (as the title and abstract imply), I recommend the authors present analyses that demonstrate PyLAE’s sensitivity and specificity. I further recommend that the authors address the following questions in the manuscript:
• How concordant are PyLAE results with results from other local ancestry programs?
• Using simulation, how accurately does PyLAE capture ancestry switches? Are breakpoints introduced by recombination between different ancestries accurately captured? Does accuracy change with segment size?
• How does PyLAE’s accuracy change with more or fewer ancestral populations?
• What is PyLAE’s runtime (include information about the processor)and how much memory does it require?
• How well does PyLAE scale (in memory and runtime) with more samples, more variants, or more ancestral populations?
4. Figure 4. Please provide an in-depth caption to describe what is represented in the heatmap. I suggest at a bare minimum that color scale be provided with some information about how values depicted in the scale are generated. Readers may additionally appreciate some form of grouping along each axis to help demarcate different super-populations (e.g., boxes around populations that belong to the same super-population).
5. Figure 3. I suggest a more informative label for the x axis, maybe something like “value for largest admixture component”. Additionally, shapes and colors are not consistent between the figure legend and points in the figure. For example, the legend indicates that blue diamonds represent individuals from European populations, but there are no blue diamonds in the figure. Instead, blue points in the figure are circles and diamonds in the figure are teal, neither of which are in the legend. This makes it difficult to interpret the figure and evaluate what is written in the text about the figure.
6. Lines 117-119. Although the number of snps in the dataset used in the manuscript suggests that linkage disequilibrium between snps may not be a substantial problem for these analyses, accounting for linkage disequilibrium between sites is critical to avoid likelihood mis-specification for researchers interested in using PyLAE. Please elaborate on the relationship between marker density and linkage disequlibrium, and if possible, provide a range of marker densities that would be acceptable. Specifically, since this work was performed in humans, what is an acceptable range of marker densities or how much LD can be tolerated? Also, what does it mean for a distance between two points to be “above LD” (lines 118-119; 132)? Does this refer to sites being in approximate linkage equilibrium?
Minor Points:
7. Figures 1 and 2. The captions of both Figures 1 and 2 mention brown as a color in the plot, but I do not see a color that appears brown in the figure. I suspect that the brown mentioned in the caption corresponds to what appears to be orange on my screen. Please double check that all population colors are labeled appropriately.
8. Lines 239-241. Do these sentences refer to the Toscani individuals?
9. Lines 269-270. Could this frequent misclassification be the result of the supervised ADMIXTURE run not being appropriate to separate individuals in these West African populations from each other?
10. Lines 133-138. Font sizes appear to be inconsistent in this area.
11. Lines 193-195. This appears to be a duplicate sentence.
12. Lines 302-303. There appears to be an error correctly inserting the name of a figure. I believe that it should be referring to Figure 5, but this is a typo that must be fixed.
13. Lines 358-368. This paragraph seems somewhat out of place. It is definitely worthwhile to discuss how the putative ancestral populations used for analyses presented in the manuscript may have influenced the results, and to point out why the particular set of ancestral populations is appropriate compared to reasonable alternatives, however this paragraph feels more like instructions for how to use a different set of ancestral populations, and seems more appropriate for a software user-manual.
14. Abbreviations: “1kG” is used only once, but “1000 Genomes” is written many times.
15. GitHub repository: It would be helpful to provide some test data with the software so that users can ensure that their installation is working as intended.
16. I noticed on the github page that it is possible to start with data in plink format (.bed/.bim/.fam). If this is the case, how will local ancestry for a site be determined when snps are not phased?

Reviewer 2 ·

Basic reporting

The writing and logic is generally strong and clear. The topic is well-introduced and references, figures and data are appropriate.

One suggestion: A lot of discussion in the manuscript involves the differences between the “haploid mode” and “diploid mode” of the approach and it is not explained why this is so significant. It would be useful to discuss how previous approaches have dealt with the issue and/or why it is a particular problem for PyLAE.

Experimental design

The problem is clearly stated and the methodology is well-described. The PyLAE method makes sense as a solution to the problem of identifying admixture through the genome.
However, the methods used to validate PyLAE are not very convincing. The authors compare the aggregated PyLAE results (destroying the local information) to admixture vectors from Admixture (Figure 5) but this is not really a test of the accuracy of the local results because the local character of the results do not come into play here.

I appreciate that validating the local ancestry inference is challenging as we have no direct measure to compare against. However, there are multiple other approaches available for local ancestry prediction discussed in the introduction to this article and a direct comparison with PyLAE would be informative. Agreement between methods using different approaches supports all of them and disagreements would also be informative.

The authors repeatedly and believably claim PyLAE’s as a major advantage of the method but this is not demonstrated anywhere.

Validity of the findings

Aside from issues mentioned above, all conclusions are supported.

---

## Round 0.2 · Minor Revisions

Dear Dr. Moshkov and colleagues:

Thanks for revising your manuscript. The reviewers are mostly satisfied with your revision (as am I). Great! However, there are a some issues still to entertain. Please address these ASAP so we may move towards acceptance of your work.

In particular. Please consider how you can predict a significant impact of PyLAE by using comparative simulations or rationally explaining the superiority of your approach over others.

Best,

-joe

Reviewer 1 ·

Basic reporting

The revision of “Local Ancestry Prediction with PyLAE” makes it a much-improved manuscript. The authors provided point-by-point responses to most reviewer comments and have added additional information and analyses to the manuscript that were suggested by the reviewers. I commend the authors for their revisions, however after reading through the revised manuscript, I have several new comments which are detailed in the “Experimental Design” and “General Comments” sections below.

Experimental design

With the additional information provided on the difference between PyLAE’s haploid mode and diploid mode, I have some additional concerns on the validity of findings. Specifically, it appears that the difference between haploid mode and diploid mode lies primarily in how the admixture coefficients that will be used as priors are generated. In haploid mode, after phasing an individual, the prior is generated from supervised ADMIXTURE where the input data consists of homozygotic genotypes for each of the positions of the phased individual’s haplotype. I caution that such manipulation likely exaggerates differences between populations and artificially inflates (or deflates) admixture coefficients.
The addition of a comparison between PyLAE and RFMix strengthens the manuscript and the concordance of aggregated local ancestry estimates suggests that PyLAE performs well. However, the authors still do not present an adequate analysis of how well PyLAE performs locally, which is an essential element of a local ancestry algorithm. The correlation of aggregate local ancestry proportions between PyLAE and RFMix suggests that the two algorithms find the same amount of ancestry of each population, but not necessarily that they find the same ancestry at the same locations. I suggest the authors include some analysis about how concordant RFMix and PyLAE are at local scales (e.g., similarity of local ancestry inference across moving windows, determining what proportion of the local ancestry inference in concordant vs discordant).
As a final note, I mention that concordance between RFMix and ADMIXTURE is important as these are well-vetted and commonly used tools for local and global ancestry inference, but PyLAE’s concordance with these tools merely demonstrates that its estimates achieve the same standard as these commonly used tools. A more appropriate test of PyLAE’s accuracy would be to test it on simulated genotype data where there is known ancestry at each position of the genome.

Validity of the findings

Given some limitations due to issues mentioned in “Experimental Design” that may change the interpretation of some results, findings and conclusions seem consistent with results.

Additional comments

1. In responding to a comment about testing PyLAE’s accuracy, the authors performed an analysis that compares aggregated ancestry information generated with PyLAE and RFMix with ancestry proportions inferred with ADMIXTURE. The analysis demonstrates that PyLAE is largely concordant with RFMIX under a range of different window sizes and readers may be interested in this information. I request that the authors include the results from the analysis in the revised manuscript, potentially as a supplemental figure.
2. As part of their response, the authors include information about the speed on PyLAE but did not include this information in the revised manuscript. Readers interested in using PyLAE for their own research would likely be interested in having this information, and I request that the authors include it in the resubmission, or even on the software’s GitHub page.
3. Labels on figure axes throughout the manuscript are quite small and are difficult to read, even when the figure is the size of an entire page.
4. Line 438: The term “locations of regions along the genome” may be misunderstood to refer to actual positions (base pair coordinates) along the genome. I suggest using an alternate term like “population origins of regions of the genome.”

Reviewer 2 ·

Basic reporting

No new comments.

Experimental design

No new comments.

Validity of the findings

The addition of a comparison between RFMix and PyLAE strengthens the support for the claims made in the manuscript, as do other revisions. However, as in the initial submission, the only method used to assess prediction accuracy is to compare the aggregated local ancestry predictions to global predictions from Admixture. As PyLAE is designed for local prediction, the predicted local ancestry structure is itself of interest, not just the aggregated results. In the RFMix paper, for example, accuracy is assessed using the agreement between prediction and simulated results at each SNP.

The idea behind PyLAE appears sound, but it is difficult to assess and compare to other methods with no data presented on the local predictions. If the authors do not want to use simulations, it would be helpful to directly compare local predictions between RFMix and PyLAE, perhaps showing how each method predicts ancestry along some example genomes and/or producing quantitative measures of overall local agreement.

---

## Round 0.3 · accepted · Accept

Dear Dr. Moshkov and colleagues:

Thanks for revising your manuscript based on the concerns raised by the reviewers. I now believe that your manuscript is suitable for publication. Congratulations! I look forward to seeing this work in print, and I anticipate it being an important resource for groups studying the identification of local ancestry in sequence datasets. Thanks again for choosing PeerJ to publish such important work.

Best,

-joe